# How to talk so AI will learn:
# Instructions, descriptions, and pragmatics

**Theodore R. Sumers**
Computer Science
Princeton University
sumers@princeton.edu

**Robert D. Hawkins**
Princeton Neuroscience Institute
Princeton University
rdhawkins@princeton.edu

**Mark K. Ho**
Computer Science
Princeton University
mho@princeton.edu

**Thomas L. Griffiths**
Computer Science, Psychology
Princeton University
tomg@princeton.edu

**Dylan Hadfield-Menell**
EECS, CSAIL
MIT
dhm@csail.mit.edu

## Abstract

Humans intuitively use language to express our beliefs and desires, but today we lack computational models explaining such abstract language use. To address this challenge, we consider social learning in a linear bandit setting and ask how a human might communicate preferences over behaviors (i.e. the reward function). We study two distinct types of language: *instructions*, which specify partial policies, and *descriptions*, which provide information about the reward function. To explain how humans use such language, we suggest they reason about both known *present* and unknown *future* states: instructions optimize for the present, while descriptions optimize for the future. We formalize this choice by extending reward design to consider a distribution over states. We then define a pragmatic listener agent that infers the speaker's reward function by reasoning about *how* the speaker expresses themselves. Simulations suggest that (1) descriptions afford stronger learning than instructions; and (2) maintaining uncertainty over the speaker's pedagogical intent allows for robust reward inference. We hope these insights facilitate a shift from developing agents that *obey* language to agents that *learn* from it.

## 1 Introduction

Imagine taking up mushroom foraging as a hobby. How would you learn which fungi are delicious and which are deadly? Learning from direct experience [1] seems risky. But how might we best learn from others? Prior work in reinforcement learning (RL) has examined a number of social learning strategies, including passive *inverse reinforcement learning* [observe an expert pick mushrooms, then infer their reward function; 2, 3] or active *preference learning* [offer an expert pairs of mushrooms, observe which one they eat, and infer their reward function; 4–6].

We posit that few humans would rely on such indirect observations. Instead, an expert guiding a foraging trip might *demonstrate* or verbally *instruct* the learner to pick certain mushrooms [7, 8]. While such explicit instruction has been a useful tool for guiding RL agents [9–12], natural language affords much richer forms of expression. For example, an expert teaching a seminar might *describe* how to recognize edible or toxic mushrooms from their features.

To model such language use, we generalize models of *reward design* [13] to linguistic communication in a linear bandit setting. Section 2 introduces this model; Section 3 formalizes instructions and

36th Conference on Neural Information Processing Systems (NeurIPS 2022).

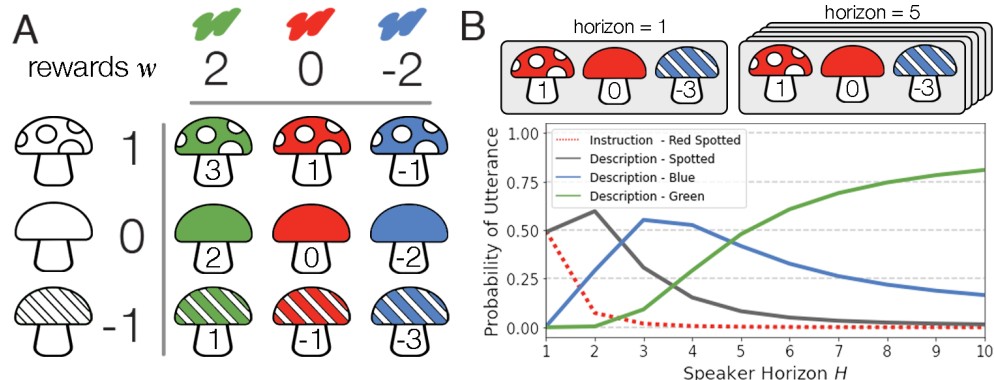

**Figure 1: A**: Rewards associated with features determine whether actions are high or low reward. **B**: Speaker's choice of utterances as a function of horizon $H$ for this start state. At short horizons (maximum supervision), speakers often use instructions or exaggerated descriptions. As the horizon lengthens, there are more unknown states, and speakers prefer truthful descriptions which provide generally useful information. Pragmatic listeners can exploit this pattern to jointly infer a speaker's horizon and reward function.

descriptions; and Section 4 defines a pragmatic listener which performs *inverse* reward design [IRD, 14], to learn about rewards from both instructions and descriptions.

## 2    Communication as reward design

**Linear bandits.** We begin by formulating the reward design problem in a *linear bandit* setting [15, 16]. Formally, we define a set of $A$ possible actions. Actions are associated with a binary feature vector $\phi : A \to \{0, 1\}^K$ (e.g. a mushroom may be green or not; have spots or not). Rewards are defined as a function of these features: $R : \phi(a) \to \mathbb{R}$. We assume they are a linear combination of the features:

$$R(a, w) = w^\top \phi(a) \tag{1}$$

so $w$ is a vector that defines the value of each feature (e.g. green mushrooms are tasty and blue are toxic; see Fig. 1A). Each task consists of a sequence of $H$ i.i.d. states. At each time step $t < H$, the agent is presented with a state $s_t$ consisting of a subset of possible actions: $s_t \subseteq A$ (e.g., a particular mushroom patch). They choose an action $a \in s_t$ according to their policy, $\pi_L : S \to \Delta(A)$.

While the bandit problem is typically considered as an individual learning problem, we assume that rewards are not directly observable and instead ask how agents should learn *socially*. We formalize the social learning problem by introducing a second agent: a speaker who knows the true rewards $w$ and the initial state $s_0$, and produces an utterance $u$. The listener updates their policy to $\pi_L(a \mid u, s)$ before beginning to choose actions. Intuitively, the horizon $H$ determines how much supervision the speaker exerts. $H = 1$ is maximum supervision (i.e. guided foraging), whereas $H \to \infty$ is minimal supervision (teaching the listener to forage in future settings). We first assume $H$ is known to both listener and speaker, but later relax this assumption.

**Speakers as reward designers.** Drawing on the Rational Speech Act framework [RSA, 17], we define a speaker $S_1$ that chooses utterances $u$ according to a utility function $U_{S_1}(\cdot)$:

$$S_1(u) \propto \exp\left(\beta_{S_1} \cdot U_{S_1}(u)\right) \tag{2}$$

where $\beta_{S_1}$ is the speaker's soft-max temperature. But what utility is appropriate? Rather than defining utility simply as Gricean informativeness [18], i.e. inducing true beliefs, we suggest that a cooperative speaker should *maximize the listener's rewards*, thus grounding utility in terms of the listener's subsequent actions. When the state is known, the *present* utility of an utterance is the expected reward from using the resulting policy to choose an action in that state:

$$U_{\text{Present}}(u \mid s, w) = \sum_{a \in s} \pi_L(a \mid u, s) R(a, w) \tag{3}$$

This formulation is equivalent to the *reward design* objective [13, 14], where the reward designer chooses a proxy reward for a single, known MDP. However, because only the first state is known,

we must also consider how well the policy *generalizes* to other mushroom patches. Thus, unlike the reward design objective, speakers may reason about future states. We represent the *future* utility of an utterance with respect to some distribution over states $P(s)$:

$$U_{\text{Future}}(u \mid w) = \sum_{s \in S} U_{\text{Present}}(u \mid s, w)P(s) \tag{4}$$

Because states are i.i.d. in the bandit setting, a speaker optimizing for a horizon $H$ can be defined as a linear combination of Eqs. 3 and 4:

$$U_{S_1}(u \mid w, s, H) = U_{\text{Present}} + (H - 1)U_{\text{Future}} \tag{5}$$

where $H = 1$ reduces to Eq. 3. We next define how utterances may affect the listener's policy.

## 3   Choosing optimal utterances

We formally define two classes of utterances, *instructions* and *descriptions*, by specifying how they affect the policy of a "literal" listener. We then show how varying the horizon H systematically affects the speaker's choice of utterance.

**Formalizing instructions.** Instructions map to specific actions or trajectories [19, 20]. Given an instruction, a literal listener executes the corresponding action. If the action is not available, the listener chooses an action randomly:

$$\pi_{L_0}(a \mid u_{\text{instruction}}, s) = \begin{cases} 0 & \text{if } a \notin s \\ \delta_{[\![u]\!](a)} & \text{if } [\![u]\!] \in s \\ \frac{1}{|s|} & \text{otherwise} \end{cases} \tag{6}$$

where $\delta_{[\![u]\!](a)}$ represents the meaning of $u$, evaluating to one when utterance $u$ grounds to $a$ and zero otherwise.[1] An instruction is a *partial policy*: it designates the correct action in a subset of states.

**Formalizing descriptions.** Rather than mapping to a specific action, descriptions provide information about the world [21–23]. Following Sumers et al. [24], we assume that descriptions provide the reward of a single feature, similar to feature queries [6]. Formally, we define descriptions as a tuple: a one-hot binary feature vector and a scalar value, $\langle \mathbb{1}_K, \mathbb{R} \rangle$. These are messages like $\langle \text{Blue}, \text{-2} \rangle$. Given a description, a literal listener "rules out" inconsistent hypotheses about reward weights $w$:

$$L_0(w \mid u_{\text{description}}) \propto \delta_{[\![u]\!](w)}P(w) \tag{7}$$

where $\delta_{[\![u]\!](w)}$ represents the meaning of $u$, evaluating to one when $u$ is true of $w$ and zero otherwise. Intuitively, descriptions set $L_0$'s beliefs about the reward of a single feature without affecting others. Descriptions need not be accurate; for example, $\langle \text{Spotted}, \text{+2} \rangle$ is a false but valid utterance. The listener then marginalizes over possible reward functions to choose an action:

$$\pi_{L_0}(a \mid u, s) \propto \exp\{\beta_{L_0} \cdot \sum_w R(a, w)L_0(w \mid u))\} \tag{8}$$

where $\beta_{L_0}$ is again a softmax optimality.

**Horizons and utterance preferences.** We use simulations to explore the effects of speaker horizons and utterance sets. Fig. 1A shows our bandit setting. "Instruction" utterances correspond to the nine actions. "Description" utterances are the 6 features $\times$ 5 values in $[-2, -1, 0, 1, 2]$, yielding 30 feature-value tuples. We assume the listener begins with a uniform prior over reward weights and set $\beta_{L_0} = 3, \beta_{S_1} = 10$. We use states consisting of three unique actions, giving 84 possible states.

To quantify how the horizon $H$ affects the generalization of the listener's policy, we repeat the task for all 84 start states using horizons ranging 1-10 and different utterance sets. Fig 1B shows one example, and Fig 2 plots a literal listener's average future rewards. When the horizon is short (small $H$), speakers focus on the visible state, producing utterances which generalize poorly (low future rewards). As $H$ increases, they provide more generally useful information. Finally, instructions are most useful at short horizons; speakers with access to descriptions use them exclusively when $H > 2$.

---

[1]We assume that groundings are known, i.e. the literal listener understands the meaning of utterances.

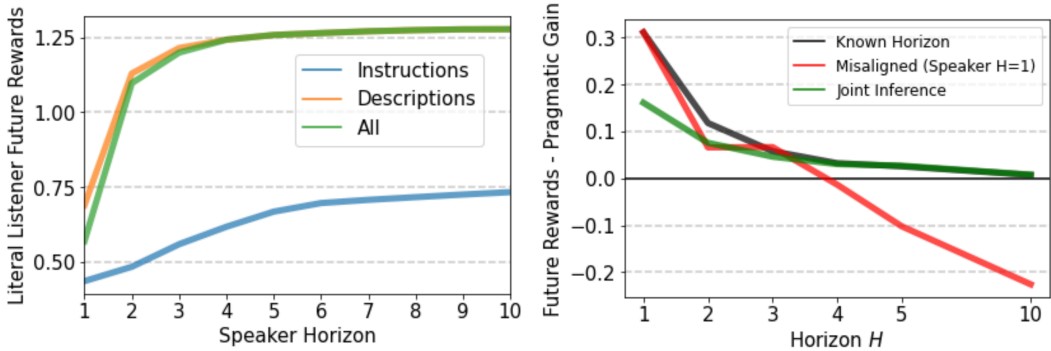

**Figure 2:** Left: Speaker behaviors, showing "future" rewards (Eq. 4, averaged over all 84 start states) for a literal listener. At longer horizons, speakers that can produce descriptions achieve high reward. Right: Listener inference, showing "future" reward gain from pragmatic inference (Eq. 4, $L_1 - L_0$ averaged over all 84 start states). Inference works best when the listener knows the speaker's horizon, but misspecification reduces performance. Jointly inferring the rewards and horizon (Eq. 10) mitigates this risk.

## 4   Learning from utterances

We now ask how the listener should *learn* from the speaker's utterance, using pragmatic inference to recover information about the reward function.

**Known horizon.** Following the standard RSA formulation, we can define a pragmatic listener $L_1$. When the speaker's horizon $H$ is known, this is equivalent to inverse reward design [14]:

$$L_1(w \mid s, u, H) \propto S_1(u \mid w, s, H)P(w) \tag{9}$$

Given an instruction, $L_1$ can recover information about the reward weights; given a description, $L_1$ can recover information about features that were not mentioned. The $L_1$ listener then chooses actions with respect to this posterior by substituting it into Eq. 8. Fig. 2 shows the gain in "future" rewards for a pragmatic listener ($L_1$ - $L_0$) when the speaker has access to both instructions and descriptions, and their horizon is known. Pragmatics are particularly helpful when the speaker has a short horizon and is *not* attempting to provide general information.

**Misaligned horizons.** Unlike IRD, in linguistic communication the speaker's horizon $H$ is not explicitly known. Prior work has highlighted the risks of assuming a human is behaving pedagogically when they are not [25], so we test this form of misalignment. We fix the speaker $H = 1$, and vary the listener's $H$ from 1-10. Fig. 2 confirms that this results in worse performance than a literal listener.

**Inference over speaker horizons.** To mitigate the risk of horizon misalignment, we can instead assume the speaker's horizon is unknown. Given an utterance, the listener jointly infers both their horizon and rewards, then marginalizes out the horizon:

$$L_1(w \mid s, u) \propto \sum_H S_1(u \mid w, s, H)P(H)P(w) \tag{10}$$

We test a pragmatic listener with a uniform prior over $H \in [1, 2, 3, 4, 5, 10]$. This results in more conservative reward inference, but avoids misalignment risk. Fig. 2 shows the results.

## 5   Discussion

In this work, we formalized communication as reward design, allowing us to unify instructions and descriptions under a single objective. Simulations show that instructions are optimal when the state is known, but descriptions are optimal when considering a distribution over states. Finally, a pragmatic listener can jointly infer the speaker's horizon and reward function. One important limitation of the current work is the assumption that groundings are known. In order to move beyond our toy setting, it will be necessary to develop methods to ground more abstract descriptive language directly to reward functions [22, 23, 26].

## Acknowledgments and Disclosure of Funding

We thank Rachit Dubey, Karthik Narasimhan, and Carlos Correa for helpful discussions. TRS is supported by the NDSEG Fellowship Program and RDH is supported by the NSF (grant #1911835). This work was additionally supported by a John Templeton Foundation grant to TLG (#61454) and a grant from the Hirji Wigglesworth Family Foundation to DHM.

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
