# OpenReview forum: "How to talk so AI will learn: instructions, descriptions, and pragmatics"
_NeurIPS.cc/2022/Workshop/LaReL — LaReL 2022_

### Official Review · Reviewer_Mrod · 2022-10-06
**Interesting proposal for reward learning from language through pragmatic inference**

**Rating:** 7
**Confidence:** 4

**Review:**

This work suggests to use language instead of numerical reward in a problem where exploration might be risky. Humans use language to specify desires and beliefs, but we do not have models that can properly do this yet. To this end, the authors consider a contextual bandit setting with a social teacher that gives descriptions or instructions. The authors use total future reward as the utility function in the RSA framework, meaning the speaker in the framework optimises a literal listeners future reward. The RSA framework now provides a computational model of specifying desires/beliefs through language as well as inferring them from language. This is an original and interesting idea. Additionally, the work is clear and presents a lot of information in limited space. The authors show when instructions are preferred over descriptions in their setup, and that a pragmatic listener that assumes information about unmentioned features can get higher reward than a literal listener. All and all, this is an interesting step towards the important goal of using pragmatics to infer rewards from language.

Pros:

- adjustment of RSA framework for RL
- this work takes step towards the important goal of pragmatically inferring rewards from language descriptions
- show that jointly inferring unknown speaker information mitigates risk of assuming the speaker is helpful

Cons:

- limited notion of language and grounding is assumed known (from instruction/description to reward function), but the authors note this themselves, it is a con but can be addressed in future work
- assumes speaker knows true reward, I think this setting is especially interesting if the speaker themselves don't even know the exact reward (could perhaps be tested in the same way as in the misaligned horizons section)
- setting is still very toy, many things to change and think about to make this actually useful in real-world RL problems (how does moving on from bandits to full RL problem affect this; real language is more noisy than this and doesn't map 1-1 onto rewards, but is rather a description of some presumably personal noisy perception of a preference: "I really like this mushroom")


Unimportant but:

- Typo line 30 s/descrioptions/descriptions
- Typo line 51 s/temperature.c/temperature.

---

### Official Review · Reviewer_gco5 · 2022-10-17
**Insightful work**

**Rating:** 7
**Confidence:** 3

**Review:**

This work investigates how artificial agents can learn reward functions from linguistic inputs.
The authors propose an original conditional bandit setting where a listener agent has to collect tasty mushrooms and avoid deadly ones given an utterance communicated by a speaker agent. In this setting, the authors use the Rational Speech Act framework to model the behavior of the speaker and listener. They formally define two communication protocols (instructions and descriptions). Their experiments clearly illustrate when descriptions are preferable over instructions (when agents need to reason over distributions of states).

I think that this paper tackles an interesting and important problem. However, the experiments are conducted in a toy setting without grounding (acknowledged by the authors). I think that it can serve as a solid basis to investigate more ambitious sequential decision-making environments potentially, with meaning negotiation between agents.

This paper is a good fit for this workshop.

---

### Decision · Program_Chairs · 2022-10-20

Accept